**www.cambridge.org/ext**

## Overview Review

law; policy; regulation; best practice; good governance

**Corresponding author:**
Katie Woolaston;
Email: katie.woolaston@qut.edu.au

# Best practice mechanisms for biodiversity conservation law and policy

Callum Brockett[1], Katie Woolaston[1] , Felicity Deane[1], Fran Humphries[1], Ethan Kumar[1], Amanda Kennedy[1] and Justine Bell-James[2]

[1]Faculty of Business and Law, Queensland University of Technology, Brisbane, QLD, Australia and [2]TC Beirne School of Law, University of Queensland, St Lucia, QLD, Australia

## Abstract

Around the world, countries have introduced laws and policies designed to prevent species extinction. While there have been some success stories, overall, these laws and policies are routinely failing. Extinction rates continue to climb. However, the law is necessary to regulate the human-environment interactions that form the basis of the drivers of extinction and biodiversity loss, including land-clearing, the discharge of greenhouse gases and the introduction of invasive species. The purpose of this paper is to evaluate the literature specifically on biodiversity conservation law, to review and describe the commonalities in laws and legal systems that can be considered successful, or unsuccessful. Laws for the conservation of biodiversity form a critical component for minimising the drivers of extinction, with species extinction being an extreme outcome of biodiversity loss. We reviewed 128 publications from around the world to ascertain and synthesise best practices in law and policy that aim to protect and conserve biodiversity (herein termed 'biodiversity conservation law'). The literature demonstrated that when it comes to biodiversity conservation law, the concept of 'best practice' is elusive, and does not necessarily equate to a reversal in species decline. Further, most western countries utilise the same legal mechanisms (also known as policy tools) for biodiversity conservation, although some countries implement these laws more effectively than others. In this paper, we explore and explain several common legal mechanisms discussed across the range of literature, including species listing and recovery plans, protected area regulation, stewardship, restoration, and offset and no net loss schemes. We also explore the necessity of biodiversity and climate mainstreaming across all laws and highlight the need to engage in genuine partnerships and collaborations with First Nations communities. This paper, and the principles explored herein, should assist law and policymakers to regulate more effectively and explain to those in the conservation sciences where research should be directed to improve the science-policy interface.

## Impact statement

The principles and mechanisms explored in this paper should assist law and policymakers to regulate against extinction with greater success and explain to those in the conservation sciences where research should be directed. In particular, researchers should associate their work with the law and policy governing their area of conservation, so that more research can be directly related to the effectiveness of legal mechanisms designed to reduce extinction rates. This will help ensure that appropriate feedback loops can be implemented in governance structures.



## Introduction

The global community is in the midst of a mass extinction event, with human actions to blame (IPBES, 2019). Human behaviours are behind key threatening processes of extinction, including deforestation and other clearing of habitat, pollution, the introduction of invasive species, and the effects of climate change, including increased natural disasters (Cresswell et al., 2021). Law is a key tool for restricting these human behaviours that lead to biodiversity decline and extinction (IPBES, 2019), and around the world, national and sub-national governments have attempted to stem the tide of rising extinction rates by introducing laws and policies to address these threatening processes.

The focus of this paper is biodiversity conservation law, which we define broadly to refer to the range of legal instruments that aim to maintain or increase species diversity and ensure the sustainable and equitable use of the resources and benefits that flow from biodiversity. Within this definition, we include laws that are targeted at conserving biodiversity rather than species extinctions as we acknowledge that laws may be more effective at reducing extinctions where this

is an explicit object of legislation (see e.g., Woinarski et al., 2017). However effective biodiversity conservation laws can play a key role in reducing extinctions (see e.g., Wintle et al., 2019).

Legal obligations related to biodiversity conservation exist at an international level through treaties and conventions (e.g., Con*vention on Biological Diversity*, *Convention on International Trade in Endangered Species of Wild Fauna and Flora*), and through domestic law (at national/state/regional levels of government). These obligations exist in legislation specifically and directly designed to conserve biodiversity and prevent biodiversity loss and extinction, as well as in laws and policies that indirectly meet biodiversity conservation goals: for example, laws that govern industries whose activities commonly result in biodiversity loss, such as forestry, agriculture, fisheries, water management, land management and planning. With such a wide range of actors and industries contributing to the degradation of biodiversity, it is difficult to define an exhaustive list of laws relevant to biodiversity conservation. Rather, all laws that may have an impact on a species' status or the environment in which they exist can be categorised under the umbrella of 'biodiversity conservation laws' and are consequently susceptible to critique regarding their ability to prevent actions that ultimately could lead to extinctions. This broad spectrum of relevance, along with the multiple levels of governance at which these laws operate, results in a complex legal network susceptible to overlap, legislative gaps (McCormack, 2018), conflicts of values (Reside et al., 2017), misdirection of or inadequate resources (Woinarski et al., 2017), and a lack of proper implementation (Martin et al., 2016).

Biodiversity conservation law has a distinct role to play in preventing species extinction by embedding values and principles to guide actors and establishing and implementing mechanisms that rectify past behaviour and regulate future harmful activities, both before and after species are identified as 'at risk' of extinction. However, biodiversity conservation law generally has not been effective in reducing extinction levels (Razzaque et al., 2019), and the evidence suggests that most nations are not currently shaping their legal systems in a manner that provides species with the best chance against extinction (Santangeli et al., 2013; Martin et al., 2016; Kraus et al., 2021). Traditionally, western legal systems have been underpinned by conceptual frameworks like Ecologically Sustainable Development (Peel, 2008), which means efforts to mitigate biodiversity loss and prevent a species extinction are often balanced with competing social and economic interests. Furthermore, even where laws are drafted to specifically address species extinction (see e.g., Evans et al., 2016) and conserve biodiversity, there is often a shortfall in implementation due to factors such as a lack of resources (see e.g., Wintle et al., 2019) and insufficient targets, monitoring, enforcement, and adaptation to changing science and social circumstances (see, e.g., Woinarski et al., 2017). Whilst recognising that truly effective biodiversity conservation requires addressing these regulatory and governance factors that exist alongside law, the purpose of this paper is to provide an overview of the scholarship regarding best practices in biodiversity conservation law and policy, with particular attention to idenitifying principles and mechanisms needed to prevent biodiversity loss. To this end, we undertook a review of the literature to determine what is considered 'best practice' biodiversity conservation law, regulation or policy. This article commences by outlining the methodology we used for this literature review and the themes that were distilled from our review of 128 papers. These themes were that best practice biodiversity laws included the identification of five key principles: (1) threat-based laws, (2) target-based laws, (3) transparency and independence, (4) inclusive decision-making,

and (5) evaluation and review, and eight key mechanisms: (1) mainstreaming biodiversity and climate change, (2) indigenous collaborations and partnerships, (3) protected areas, (4) listing and recovery plans, (5) stewardship and private protected areas (PPAs), (6) land restoration, (7) offsets and no net loss schemes, and (8) legal rights of nature.

We then address the results of the literature according to these principles and mechanisms.

We anticipate that the results of this literature review will be of use to anyone involved in the formulation or review of biodiversity conservation law, including governments, law and policymakers, scientists and conservation groups.

## Methodology

To understand the state of the scholarship concerning best practice in biodiversity conservation law, we undertook a review of the published literature. We chose Google Scholar and Scopus for our literature searches to cover two of the three most extensive literature databases, with the third being Web of Science. Previous studies have found that Scopus and Web of Science cover substantially the same content (Waltman, 2016) therefore we deemed our choice of databases as sufficient to obtain an accurate picture of the breadth of literature.

Our next step was to develop a search string strategy (see Aromataris and Riitano, 2014). We formulated the following search strings and used them to search both databases:

> Biodiversity AND conservation AND "best practice" AND (law OR policy OR regulation or governance) AND (agriculture OR farming OR pollution OR ecosystem OR habitat OR threat OR urban OR infrastructure OR climate).

As there is a large body of literature on biodiversity conservation generally, we included the limiters to ensure that only literature related to law and policy was found. We included agriculture and farming as search terms to capture laws and policies that regulate these known drivers of biodiversity loss. Further, we excluded any evaluation of biosecurity laws. Although invasive species are 'particularly destructive' in all ecosystems (McCreless et al., 2016), we excluded biosecurity law and practice as these are laws that require specific principles and mechanisms for effectiveness. Once the databases had been searched, we reviewed the results for inclusion or exclusion. Our criteria for inclusion of literature within our review were: (1) publications that were peer-reviewed (articles, book chapters) as well as grey literature (e.g., reports), (2) material published in English, (3) material published during the period 2012–2022 to limit our analysis to relatively recent developments in the topic; and (4) material that focused on law and policy discussion. To this end, we excluded literature purely analysing biodiversity conservation from a scientific perspective or other disciplinary perspective, or that did not mention law or policy, or only in passing. Each database's results were reviewed by two members of the research team to negate the risk of relevant papers being omitted. Each member of the research team also made individual judgements on the relevance of literature based on the common criteria described and with the objective of understanding best practice principles in biodiversity conservation law.

To ensure that we found the most relevant literature we supplemented our searches with a 'backward snowball method' (Wohlin, 2014) where we reviewed the reference lists of articles and included relevant articles from those lists. Again, we used the criteria above and excluded articles that did not, from their titles, meet these

criteria. We also excluded articles that had already appeared on our search lists or that included a discussion of biosecurity (as this was outside the scope of the analysis). We also considered the context in which the paper was cited in the primary paper under review to assist with determining relevance (Wohlin, 2014).

Finally, if a paper was still included following the application of the inclusion or exclusion factors, we proceeded to read its abstract and determine whether to include it, with the exclusion of best practice principles being the most common eliminating factor.

After this process, we had our final list of 128 articles (see column 1 of Appendix). To support multi-author analysis of the articles, each team member was assigned a group of articles to initially read, and keywords were distilled from these articles (see Appendix). As per a conventional content analysis approach (Hsieh and Shannon, 2005) we inductively developed coding during the review by using keywords that commonly appeared. After this initial review, we then revisited these codes and ultimately determined that the relevant themes spanned five key principles and eight mechanisms (see columns 2 and 3 of Appendix).

## Results of the literature review

Our specific area of enquiry in our literature review was to determine best practice in biodiversity conservation law. Broadly, the literature demonstrated that best practice biodiversity conservation law is underpinned by two things: the incorporation of key principles, and their implementation in particular mechanisms. To be effective, a biodiversity conservation law must generally consist of both principle/s and mechanism/s. For example, a law may set a target for biodiversity conservation outcomes (a principle). However, this target is unlikely to be achieved in the absence of a mechanism to achieve this (e.g., the creation of a protected area). A law will not necessarily need to include all the principles and mechanisms identified in our literature review: rather, this will be context-dependent.

Further, a law needs to go further than simply setting out a principle or creating a mechanism. For instance, a review of the conservation literature might suggest that the ability to list 'threatened ecological communities' is an essential component of a biodiversity conservation legal framework (Dorrough et al., 2021; Kraus et al., 2021). However, simply including an avenue to list threatened ecological communities may not actually result in threatened ecological communities being listed within a given framework, and so may fail to meet a conservation objective. The policy tool or mechanism, in this case, listing processes for threatened ecological communities, needs to go beyond the science and be designed and implemented in a way that ensures the tool itself is not meaningless. Additionally, effective laws need to be designed in a way that ensures compliance (sometimes this means that public support is necessary, and there are relevant incentives for compliance) and are compatible with existing legal frameworks (such as those that regulate private property).

This section outlines the results of our literature review, grouped around the five key principles and eight mechanisms that may form part of an effective biodiversity conservation legal framework. The principles are: (1) designing laws to address threats, (2) ensuring laws are based on or embed conservation targets, (3) integrating transparency and independent decision-making and review options, (4) designing laws and legal systems that are inclusive, and (5) ensuring effective evaluation and review mechanisms. We also acknowledge other important principles of environmental laws (e.g., the precautionary principle and intergenerational equity), but

these principles did not naturally emerge in this literature review as necessary to incorporate explicitly in law and policy. Hence, we recognise this limitation.

The mechanisms are (1) mainstreaming biodiversity and climate change, (2) indigenous collaborations and partnerships, (3) protected areas, (4) listing and recovery plans, (5) stewardship and PPAs, (6) land restoration, (7) offsets and no net loss schemes, and (8) legal rights of nature.

Although the literature review revealed some differences in the analysis of these mechanisms, we suggest that within a best practice biodiversity conservation legal framework each mechanism will have a different role and function in furthering the general principles listed above. Whilst each individual law is unlikely to incorporate every mechanism, the literature suggests that best practice is characterised by a system that incorporates a range of mechanisms to ensure a cohesive approach to preventing the different drivers of species extinction.

We also acknowledge that this literature review does not provide an exhaustive synthesis of every possible mechanism, but rather it focuses on a limited list that incorporates mechanisms identified most often in the literature reviewed. Other developing mechanisms, such as extinction declarations and ecotourism, and the incorporation of these in law and policy are worthy of future analysis, but outside the scope of this paper.

## Principles found in best practice biodiversity conservation law and policy

### Threat-based laws

Our review of the literature indicated that best practice principles require the law to address threats to biodiversity (Scheele et al., 2018) with the identification of threats and the necessary actions to address them based on scientific input (Evans et al., 2016) and Traditional and local knowledges (Artelle et al., 2019). Best practice also means that threat abatement should also occur simultaneously with protection policies and laws, with processes in place to mitigate the threat source (Hutchings et al., 2016).

Our review also found that laws should adopt a scaled approach, and biodiversity conservation law and relevant policies should accommodate different species' threat levels and incorporate varying responses depending on that threat. For example, species with a small population and narrow habitat are specifically vulnerable to land use change (Sykes et al., 2020) which enhances extinction risk (Staude et al., 2020). The law should also provide for decision-makers and authorities to undertake a science-based assessment of the level of threat to a species (e.g., vulnerable, endangered) and the specific source of the threat (e.g., land clearing), with species that face a higher risk of extinction to be the focus of immediate and concerted conservation actions (Kraus et al., 2021).

Law and policy should also allow for threat-based laws to adapt to changing circumstances and threats that arise in the future, adequately abating the drivers of species extinction as they evolve. For example, incidental catching of seabirds during oceanic longline fishing operations was listed as a key threatening process under the Australian *Environmental Protection and Biodiversity Conservation Act*, but successful threat abatement plans have reduced the deaths of albatross and petrels in Australia (Baker et al., 2002). Elements recognised as contributing to the success of this threat abatement plan included semi-regular updates, implementation through regulations and management plans, wide support from

stakeholders (especially government departments), criteria against which outcomes are measured, specific enforcement mechanisms, and a strong NGO advocate (Invasive Species Council and Bush Heritage Australia, 2020). The funding available to support the instrument is also likely to be critical to success (Bottrill et al., 2011).

### Target-based laws

Our review identified that one of the most important features of best practice biodiversity conservation is the inclusion of goals, targets and indicators within biodiversity conservation laws (Maseyk et al., 2019). Appropriate and explicit targets and indicators allow the effectiveness and efficiency of the law or policy to be measured, as well as ensure accountability for the success of prevention measures (Scheele et al., 2018). Targets should communicate clear benchmarks for assessing whether objectives, principles or rules around a legal biodiversity mechanism have been met, which leads to further evaluation and adaptation (Evans et al., 2016). While specific targets and indicators may vary depending on a range of circumstances, biodiversity laws can still legislate avenues for these targets and indicators to be established. For example, the Australian state of Victoria has legislated a requirement that a Biodiversity strategy is prepared, which must include targets to measure achievements (*Flora and Fauna Guarantee Act 1988* section 17), without legislating the specific targets in the law itself.

Broad targets and indicators have been proposed or implemented on an international scale, which can be reinforced in domestic or national law. The proposed Post-2020 Global Biodiversity Framework is an example of this, providing a framework for the development of national and regional goals and targets that further the objectives of the *Convention on Biological Diversity* ('CBD') (CBD Working Group, 2021). A biodiversity framework that incorporates an integrated system of goals, targets, and indicators is a best practice approach to biodiversity conservation. For example, Canada's government released the *2020 Biodiversity Goals and Targets for Canada* in response to global Aichi Biodiversity Targets. These included a range of quantitative and qualitative targets to be met by 2020, including percentages of land to be conserved through protected areas (17% of terrestrial areas and inland water and 10% of coastal marine areas protected by 2020), implementing priority adaption measures, reduction of pollution levels, and Aboriginal traditional knowledge regularly, meaningfully, and effectively informing biodiversity conservation and decision-making processes (Environment and Climate Change Canada, 2016).

### Transparent and independent decision-making and review

The decision-making process in laws that aim to prevent species extinction should reflect transparency, accountability, and participation (van Doeveren, 2011). Laws should ensure that government and ministerial decision-making processes are scrutinised by independent review, with triggers in place for intervention in ineffective management processes. Independent or quasi-independent scientific authorities (such as an Environmental Protection Authority) may minimise ministerial discretion and prioritise evidence-based policy over political influences (Department of Environment and Science (Qld), 2022). Decision processes prescribed by law should be based on empirical scientific data and understanding, and be informed by independent scientific authorities (Evans et al., 2016; Dorey and Walker, 2018; Bethlenfalvy and Olive, 2021). For this best practice principle to be reflected in law and policy, provisions should limit the discretion of decision-makers to carry out their functions based on social and economic factors that do not prioritise species extinction prevention.

The literature also suggests that biodiversity conservation law and policy should be transparent and made publicly available (Richardson, 2016; Scheele et al., 2018; Hilty et al., 2020). This includes any law or decision-making process, as well as any monitoring, evaluation, triggers for intervention, costs and burdens, and terminology. Publication of these elements should also be accompanied by public consultation and review processes. Public consultation ensures participation in the decision-making process, with open invitations to the public to make submissions on legal mechanisms that aim to prevent species extinction (with accompanying provisions that require the decision-maker to properly consider these submissions). Best practice is also indicated by law that enables avenues for the review of these decisions, not only for those immediately affected by a decision (e.g., those prevented from performing an activity) but also instilling broader provisions for reviews for a wide range of parties who have an interest in preventing species extinction (Tsioumani, 2018).

Effective biodiversity conservation law should also be backwards-looking, and it has been argued that where a species extinction does occur there should be a public inquiry to examine the causes and actions that contributed to it (Woinarski et al., 2017; see also McCormack in this special issue). This can ensure accountability and review of previous decisions, and ensure that future decisions are informed by any identified shortcomings (Smith et al., 2018).

### Inclusive decision-making

Best practice biodiversity conservation law should incorporate the principle of inclusive decision-making. Whilst this overlaps to an extent with public consultation in transparent decision-making as discussed above, inclusive decision-making as a distinct principle expands upon the former by reinforcing the need for laws to include all relevant members of the public or specific groups involved in decisions that affect the environment. Public participation is reflected in human rights law, for example, the inclusion of the right to effective participation of a minority population as part of the right to culture (Human Rights Committee, 1994). Public participation promotes democratic values within a governance system, with benefits including stakeholder empowerment, as they can influence matters that affect them, which leads to greater community cooperation and an increase in behavioural change, including a greater conservation ethic (Cattino and Reckien, 2021; Hao et al., 2022).

In the specific context of biodiversity conservation law, the literature suggests that best practice biodiversity conservation laws and policies should develop and maintain Indigenous collaborations and partnerships. These partnerships extend beyond mere participation and require the recognition of Indigenous authority in traditional ecological knowledge systems (Artelle et al., 2019). Literature suggests that biodiversity law and policy should not limit Indigenous rights and access to land and should aim to enhance the position of Indigenous peoples through the preservation of language and culture (Berkes and Davidson-Hunt, 2006; Reimerson, 2013; Satterfield et al., 2013). This principle is further discussed below.

The literature also notes that best practice inclusive decision-making in biodiversity law is also reflected in effective efforts to

develop and maintain partnerships between private landholders and regulators (Langpap et al., 2018; Lee and Wakefield-Rann, 2021; Pannell et al., 2021).

### Effective evaluation and review

Best practice biodiversity law and policy should include provisions that allow for effective evaluation and review of the law itself, as well as any function, activity, or program carried out under the law (Verschuuren et al., 2021). Measuring the effectiveness of biodiversity mechanisms, through monitoring and evaluation, results in laws being adapted to changing circumstances, new scientific findings, and any ineffective management processes identified through the evaluation process (Hutchings et al., 2016). Monitoring and evaluation are integral to the effectiveness of international biodiversity frameworks. This is essential to provide baseline data against which to determine 'effective' implementation, promote accountability, track the progress of the conservation targets and promote policy and decision-making that is informed by evidence (Mascia et al., 2014; Cardesa-Salzmann, 2016). The literature also suggests that laws must also provide timely and public systems of review to the law and any programs carried out under the law, with monitoring and evaluation informing this review process. Operating in conjunction with established targets and indicators, monitoring and review programs are outlined in the proposed Post-2020 Global Biodiversity Framework (CBD Working Group, 2021). Legislated dates for the review of frameworks, as well as provisions for adaptive management principles, can assist with provisioning for this principle (Evans et al., 2016; Dimitropoulou, 2018; Scheele et al., 2018).

## Mechanisms found in best practice biodiversity conservation law and policy

### Climate and biodiversity mainstreaming

One prominent theme that emerged from the literature was that biodiversity conservation requires mainstreaming in law and policy. Mainstreaming can be achieved through different means but in general, it is supported when a particular issue, in this case, climate change and biodiversity conservation, becomes a core consideration of the government, and government processes and systems are redesigned and re-organised from the perspective of addressing those issues (Forester and Bleby, 2022). Mainstreaming in biodiversity law – also referred to as horizontal policy integration (e.g., Lafferty and Hovden, 2003) – mainly involves placing one issue more centrally on the agenda of another domain. For example, species extinction prevention becomes a core concern of the legislation governing industries that typically affect biodiversity (i.e., forestry, agriculture, fisheries, water management, land management and planning) (Karlsson-Vinkhuyzen et al., 2017). The literature diverges on the extent to which mainstreaming must be implemented in law and policy to constitute 'best practice'. The argument that environmental factors must take priority over other concerns to constitute mainstreaming (Lafferty and Hovden, 2003) is contrasted with the notion that mainstreaming involves the process of merging the concerns of two domains (Jordan and Lenschow, 2008). As a best practice approach to mainstreaming, legislation should include an intention to make species extinction prevention a core priority of the law, forming a primary and compulsory directive or guiding principle, rather than only one factor to be considered. Social, cultural or economic considerations should be ancillary to the main goal of preventing species extinction to constitute biodiversity conservation mainstreaming (Tallis et al., 2015; Maseyk et al., 2019). Threat and target-based laws, with objective criteria that are underpinned by scientific evidence and scrutiny, can ensure species extinction is prioritised over social or economic influences, with only very limited exceptions where environmental or social justice requires it. This means biodiversity and environmental concerns must not only be present in the broad objectives and principles of law, but also in the specific functions of authorities under the act.

Any function or activity which further prioritises species extinction prevention may constitute 'mainstreaming', with no exhaustive list of actions that may evince 'mainstreaming' and further the above principles. For example, mainstreaming may include the establishment of a central climate or biodiversity conservation agency that other departments will report to, reflecting structural integrity principles through enhancing coordination between government agencies for decision-making and reducing the cost of conservation bureaucracy (see, e.g., the *Climate Change Act 2017* in the Australian state of Victoria). Other examples range from introducing compliance reporting, and developing Ministerial guidelines that direct proper consideration of species status in decision-making, to the inclusion of key performance indicators in executive contracts, performance plans and training modules – all of which reflect effective evaluation and review principles (Forester and Bleby, 2022). Mainstreaming can be furthered through a range of other, specific mechanisms, and as a result of this scope, reflect and further a range of best practice principles.

### Indigenous collaborations and partnerships

The literature was clear that biodiversity law and policy that provides avenues for Indigenous collaborations and partnerships reflect inclusive decision-making best practice principles. However, biodiversity law and policy that promotes these collaborations should aim to balance conservation goals and human rights goals (i.e., the rights of recognition, and well-being of Indigenous peoples) to ensure mutual benefit for both parties (Artelle et al., 2019). Recognising Indigenous ecological knowledge can empower Indigenous peoples through joint decision-making (Berkes and Davidson-Hunt, 2006), as well as contribute to recovery from colonial assimilation, state violence, and ecological degradation (Satterfield et al., 2013). In furthering conservation efforts, Indigenous peoples have an in-depth knowledge of land or place and are often well-positioned to provide effective monitoring and evaluation in biodiverse areas, with traditional knowledge informing biodiversity assessment and conservation priorities (Ban et al., 2014; Artelle et al., 2019).

These collaborations and partnerships, as a best practice mechanism, must recognise the authority and rights that Indigenous people can have about preventing species extinction while ensuring these partnerships benefit the position of Indigenous peoples (Goolmeer et al., 2022b). The Indigenous Land and Sea Management Programs funded by the Australian Federal Government are examples of these goals being balanced. Such programs aim to advance biodiversity conservation and protect natural resources while also creating substantial employment and economic opportunities for Indigenous people, as well as improving the well-being of these communities through a wide range of social and cultural benefits (Larson et al., 2020). Best practice Indigenous collaborations should also extend beyond rights to natural resources and land management in traditionally owned areas, but also incorporate

traditional ecological knowledge in assessment processes and combine with western sciences to underpin decision-making processes.

### Protected areas

Empirical evidence has suggested that identifying an area as protected can provide additional protection for biodiversity conservation. However, legislative provisions that establish and provide for the management of these protected areas must incorporate certain aspects to constitute best practice, including broad representation, size and connectivity and co-existence with other laws including species listing (Mitchell et al., 2018).

Reflective of the inclusive and transparent decision-making principles, input into the nomination of areas should extend beyond government authorities and scientific bodies to NGOs and community groups (Scott, 2016). The New Zealand *Marine Reserves Act* demonstrates a particularly wide approach, with applications open to universities, private bodies, scientific research groups, Indigenous people who have *tangata whenua* status over the area, as well as the Director-General of Conservation (*Marine Reserves Act 1971* s 5; Scott, 2016). This could be broadened further by legislating that applications may be open to anyone with a special interest in nominating a protected area.

Law and policy must also recognise a need for larger and interconnected protected areas, specifically providing connectivity with existing protected areas, ecological corridors and Other Effective Area-based Conservation Measures (OECMs) (Hilty et al., 2020). Law and policy surrounding protected areas should provide avenues for interim protection measures, so that areas and the inhabiting species not yet legally listed may still be protected. For example, in New Zealand, land not designated as protected but has also not been fully assessed for its conservation value can be designated as 'stewardship land', allowing for greater or lesser protection measures once the assessment is complete (Koolen-Bourke and Peart, 2021).

The principle that requires law and policy to be designed to address threats is particularly relevant for protected area laws. Threatened species can only benefit from the establishment of protected areas if the threat abatement of pervasive threats beyond land use, such as fire patterns, exotic plants, animals and pathogens, and visitors, occurs (Taylor et al., 2011). Protected areas cannot exist in isolation to constitute best practice, but rather work in conjunction with other mechanisms to ensure these areas remain completely protected.

### Listing and recovery plans

The listing of threatened species, and associated recovery plans for these species, were identified in the literature as best practice mechanisms when law and policy incorporate specific markers. The IUCN Red List categories and criteria are an example of a best practice for the listing process. This process provides an objective criterion as well as a standardised and transparent approach to evaluating a species' status, with constant updates and improvements being implemented (Braby, 2018). Species being associated with a distinct category (i.e., extinct, extinct in the wild, critically endangered) is reflective of the threat-based law principle, recognising the level of threat of each species based on population size reduction, geographic range, and other science-based criteria (Moir and Nand Brennan, 2020). Using empirical scientific data as the basis for categorising a species was a prominent aspect of best practice, limiting considerations for socio-economic considerations

and political influence (e.g., not listing a species as it may impact economic growth for a certain primary industry) (Dorey and Walker, 2018).

Other markers for best practice in listing include the need to list ecological communities, as well as individual species, broadening the impact listing processes can have (Kraus et al., 2021). Law that provides for the listing processes to identify and protect critical habitats is also considered an element of best practice, as is evident in the United States' *Endangered Species Act* requiring that a listed species have a designated habitat (Henson et al., 2018). Law that considers listing of critical habitat and ecological communities allows for recovery to address threats to an area rather than a specific species (Walsh et al., 2013; Braby, 2018; Kraus et al., 2021), which is reflective of law that is threat-based and serves to further this principle in the best practice mechanism. Walsh et al., argued that the concept of a national threatened species list, rather than specific state or provincial lists improves the listing processes' efficiency, knowledge and data-sharing capabilities and effectively utilises limited funding (Walsh et al., 2013), although we acknowledge that this may also be contingent on this national list being accompanied by implementing legislation. Implementing this would further the structural integrity of the mechanism, with consolidated legislation ensuring more efficient coordination of resources and reducing any overlap or unnecessary bureaucracy that delays urgent action (although there may be discrepancies in a species' level of risk in one area vs. another).

The literature described a range of best practice markers for the recovery plan mechanism. These included (Dee Boersma et al., 2001; Li et al., 2020):

- legal avenues for adaptive plans that adjust to new information or changed conditions, with changes based on systematic monitoring and review;
- legal requirements for implementation of monitoring; target-based laws that allow for timely and appropriate goals to be set and defined;
- single-species focus to be balanced with multi-species and ecosystem-based recovery to ensure effective analysis is not sacrificed for efficiency;
- and laws that strengthen inclusivity through non-government participation in the recovery of threatened species.

Recovery planning should also incorporate threat-based laws as a best practice principle, as recovery is likely to be most effective when aiming to ameliorate primary threats driving decline (Woinarski et al., 2017). In the past, recovery measures have failed to prevent species extinction in instances where primary threats to a species were not identified and not subject to threat-abatement measures, rendering recovery ineffective (Woinarski et al., 2017). The US *Endangered Species Act* ('ESA') implements some of these factors, including recovery plans that estimate the time and cost of recovery, descriptions of location-specific recovery actions, and criteria for recovering a species. However, recovery measures under the US System do not require an explanation of the causes behind a species' decline, hindering the ability to mitigate the drivers of species extinction (Li et al., 2020).

Other issues with listing and recovery processes have also been raised, and these can indicate what best practice should incorporate. Some literature refers to a bias in these listing mechanisms, with less charismatic species receiving little attention regardless of their status (Dorey and Walker, 2018). Walsh et al. also identified the problem of bias in the listing of more charismatic species and noted that this may overlook some that are nevertheless vital for

ecosystem function such as invertebrates, plants and fungi (Walsh et al., 2013). Some literature also identified a lack of accountability for or allocation of responsibility for the implementation of recovery measures (Woinarski et al., 2017).

### Stewardship and PPAs

Stewardship, while varying in its formality (ranging from 'handshake agreements', to formalised programs) and PPAs, can generally be characterised by collaboration with private landowners, with efficient inclusivity a key principle in determining the best practice of this mechanism. Lee and Wakefield-Rann argue that this collaboration mechanism is a high priority due to the significant portion of natural resources located on private land, with the accessibility of this land and permissible management practices largely determined by landowners (Lee and Wakefield-Rann, 2021). For example, the effectiveness of the US ESA is largely attributed to provisions that foster partnerships between government and private landowners, implementing conservation on private land through 'Habitat Conservation Plans' (Langpap et al., 2018; Kraus et al., 2021).

The literature revealed that engaging landowners in species extinction prevention on private land is dependent upon close-working relationships as well as clear and accessible information on the benefits and costs of conservation (Pannell et al., 2021). Issues in the design and implementation of PPAs and stewardship arise due to the voluntary nature of this mechanism, relying on the willingness of landowners to participate in any conservation programs (Pannell et al., 2021). Embedding these 'soft law' programs in legislation can allow for greater institutional support. The mutual benefits of these mechanisms must be clearly communicated. Permits under the ESA enable landowners to develop lands inhabited by endangered species contingent on an approved Habitat Conservation Plan (HCP), which aims to mitigate harm to that species, while the 'Conservation Banking' programs allow landowners to protect habitat for endangered species and then sell conservation credits to developers to mitigate habitat alteration (Evans et al., 2016). The 'Safe Harbour' and 'Candidate Conservation Agreements with Assurances' programs also allow landowners to gain assurance that their voluntary actions to improve habitat or increase species numbers will not result in additional regulation (Evans et al., 2016). While limited in its effect, best practice of this mechanism is dependent on transparency and appropriate engagement with stakeholders, balancing compensation payments and regulatory assurances with conservation benefits (Evans et al., 2016). The 'voluntary' nature of these mechanisms means that 'best practice' generally requires that other actions, such as direct acquisition and protection of private land by conservation managers, are encouraged (Kearney et al., 2022).

### Ecosystem restoration

While the term 'ecosystem restoration' may cover a broad range of practices occurring in terrestrial spaces, there were general markers of best practice identified in the literature. This includes establishing legislative frameworks with clear and consistent terminology, including statutory definitions, to ensure greater public accessibility that enhances accountability (Nzyoka et al., 2021). Clear and appropriate statutory goals should also be evident in the legislation, to ensure efforts are focused, targeted, and can be monitored to measure success (which can lead to adaption if deemed to be unsuccessful) (Gann et al., 2019; Nzyoka et al., 2021). Ecosystem restoration as a best practice mechanism also requires avenues for

public participation and stakeholder support (Richardson, 2016; Campbell et al., 2017; Gann et al., 2019). This is a result of ecosystem restoration programs displacing people, which risks generating opposition that may hinder success (Richardson, 2016). Negating this effect relies on increasing social acceptability, which may involve legal avenues for offsetting disproportionate costs for a community or financial compensation, as well as sharing of experience and expertise (Richardson, 2016; Gann et al., 2019). Implementing clear and consistent target-based laws, inclusive decision-making, and furthering general social acceptability is essential to garnering support for the demanding workload involved in ecosystem restoration. The literature also raised the dependency ecosystem restoration projects have on financial capital through public funding, with the suggestion that tax law is 'reworked to stimulate philanthropic commitments while business law may overcome barriers to commercial approaches to eco-restoration' (Richardson, 2016; Campbell et al., 2017).

There are examples of these principles in ecosystem restoration projects. In the US, the *Omnibus Public Land Management Act 2009* defines the specific 'environmental baseline' that the Collaborative Landscape Restoration Program aims to restore (the structure and composition of old-growth stands according to pre-fire suppression old growth conditions characteristic of the forest type), as well as outlining general ecological outcomes (improving fish and wildlife habitat, water quality, and preventing invasions of exotic species) (Richardson, 2016). In contrast, statutory goals and targets under New Zealand law are broad and unspecified (i.e., 'restoring and enhancing the area and its heritage features, restoring and enhancing degraded landscapes' (Waitakere Ranges Heritage Area Act 2008)). However, there are examples of local community participation in New Zealand restoration projects, with Māori and environmental groups establishing individual 'eco-sanctuaries', as well as private sectors shouldering the financial burden in the wake of budget cuts in the public sector (Richardson, 2016).

### Offsets and no net loss schemes

The literature we reviewed was predominantly critical of offset and no net loss schemes (NNL), suggesting that offsets should only be used as a last resort by developers to compensate for unavoidable damage (Bull et al., 2013; Simmonds et al., 2020a; Schoukens and Van Hoorick, 2021). This is because offsets have suffered from persistent implementation failures due to weak compliance or regulatory enforcement, inconsistencies with governance arrangements (Zu Ermgassen et al., 2019), high costs for participation, and administrative delays (Martin et al., 2022), as well as NNL policies being subject to the influence of uneven power dynamics or vested interests (Zu Ermgassen et al., 2019). As a result, environmental markets are largely viewed as not adequately addressing socio-ecological issues (Lockhart and Rea, 2019), with minimal regard for their practicality (Zu Ermgassen et al., 2019). This suggests that offsets and NNL schemes may not be considered a best practice mechanism in biodiversity law and policy.

However, offsets and NNL policies are becoming increasingly relied upon by countries to mitigate the impact developments have on biodiversity (Gardner et al., 2013; Zu Ermgassen et al., 2019). This means recognising that potential best practice markers are important to develop and improve biodiversity offset outcomes. To understand, define and action these markers more research is needed and should be prioritised in circumstances where regulators wish to introduce offset policies. Examples of these best practice

aspects include the need for offsets to always be accompanied by a no-net loss policy to be effective, the adoption of a precautionary approach to offset design, access to adequate data and technical expertise, economic and financial safeguards, and strong monitoring and enforcement, as well as outlining strict and explicit thresholds where impacts cannot be offset (Bull et al., 2013; Gardner et al., 2013; Benabou, 2014; Bergman et al., 2020).

### Legal rights of nature

The literature recognised that both property rights and the sustainable development paradigm (or the misunderstanding of it) have resulted in a disproportionate emphasis on positive economic outcomes, leading to widespread environmental degradation (Riley, 2019). Many commentators argued that adopting either an environmental personhood approach or recognising legal rights for the environment would be an effective means for overcoming these problems. The differences between the two approaches can be subtle, but important when considering how they can support genuine conservation. A limitation of the rights approach is that the protection offered by providing legal rights will be limited to the specific rights that are granted to either the environment or nature in general or to distinct ecosystems (Riley, 2019). For instance, the rights granted could potentially include the right to restoration for any actions that cause environmental damage (Athens, 2018). The alternative, which is recognising 'legal personhood' for the environment means that 'an entity exists for its own interests and not for the value it contains for others' (Athens, 2018). There are several practical examples where jurisdictions have extended legal rights to the environment. The recognition of environmental legal personhood has been both sporadic and jurisdiction-specific however, which makes it difficult to evaluate the effectiveness of this as a legal mechanism to support conservation (Putzer et al., 2022).

The first constitutional recognition of rights to the environment was in 2008 when Ecuador passed a constitutional amendment acknowledging the rights of nature (Athens, 2018). However, arguably a more effective (but less general approach) was taken in New Zealand when the Parliament passed the *Te Urewera Act* in 2014. This Act embraced the traditional values of the Māori people in support of adopting legal personhood for Te Urewera national park and the Whanganui River. In this instance, the rights were attached to particular ecosystems but gave those ecosystems all the "rights, powers, duties and liabilities of a legal person" (*Te Awa Tupua (Whanganui River Claims Settlement) Act* 2017 (NZ) s 14(1)) Overall the literature suggests that although this approach aligns with a recognition of the intrinsic value of the environment, it is by no means a panacea for biodiversity conservation (Putzer et al., 2022).

Despite recognising this, ecocentric approaches to ownership could be a useful mechanism for environmental protection laws (Hoops, 2022). Commentators considered this (and the limits of western legal systems' property rights) both for the environment and for fauna impacted by environmental degradation (Best, 2021). Indeed, some commentators argue that the legal status of animals as property has significantly increased their vulnerability in times of disaster (Best, 2021).

In terms of best practice markers, these are difficult to articulate across jurisdictions generally as different nations have different requirements for constitutional amendments and likewise may approach rights differently. The New Zealand example demonstrated that common law systems can indeed provide the flexibility to support legal personhood for the environment. Where

conservation of biodiversity can sit outside regulations (such as through constitutional recognition), a rights-based approach may indeed lead to long-term benefits.

### Conclusion

The United Nations has identified that minimising or reversing the risk of species extinction is a priority for law and policy in the coming decades (CBD Working Group, 2021). Accurate and timely assessment of the processes to prevent this extinction is of utmost importance. Our literature review has sought to provide an overview of the literature on best practice biodiversity conservation law. We have presented these findings through the best practice principles identified, and the degree to which these principles have been reflected in mechanisms necessary to support best practice biodiversity conservation.

Despite identifying these principles and mechanisms, we found that the specific concept of 'best practice' law and policy for mechanisms to prevent biodiversity loss was elusive in the literature. There are several reasons for this, including the constantly evolving nature of threats to species extinction and the corresponding mechanisms designed to prevent extinction. This difficulty is compounded by the disconnect between law and policy and the degree to which legislated measures are implemented in a practical sense (whether this is a result of limited financial resources, political influences, or social pushback and lack of enforcement in response to noncompliance). However, the literature demonstrated that when biodiversity law and policy implements best practice principles the risk of ineffective implementation is mitigated, for example, through provisions that allow for public consultation to ensure stakeholder engagement, or laws based on real and perceived threats to maximise effectiveness and efficiency. Mechanisms that aim to prevent species extinction can be considered best practice only when these core principles are reflected and enhanced in biodiversity law and policy (or any functions or programs that are carried out under these laws).

We note our literature review had some limitations. First, the jurisdictional scope was restricted as the databases we searched were focused on Western legal systems, predominantly North America, the UK, Australia, New Zealand and Europe. As such, our discussion is of best practice in western legal systems and may not necessarily reflect best practice from other parts of the world. Second, biosecurity was excluded from the analysis to limit the scope of the review. Although invasive species are a driver of extinction, it was outside the analysis of this project to consider laws with this objective. We acknowledge that this is a significant omission from our analysis and suggest it is an important topic for future research projects. Finally, consideration of best practices necessarily limited the analysis to a select body of literature. This search term meant that there were many articles relevant to biodiversity conservation law and policy that were not analysed in this review, which could have limited the principles and the mechanisms that we considered. For instance, any future project would necessarily consider some of the more relevant principles of sustainable development, such as the precautionary principle and intergenerational equity.

Despite these limitations, the findings of this review are important in terms of understanding principles and mechanisms for best practice biodiversity conservation law and policy. In particular, to ensure 'best practice' in combating the threats to species, cooperation and mutual understanding between relevant fields of understanding and knowledge are vital. Scientific research, evidence, and

input must inform the function of government decision-makers and regulators, while conservation science must understand how law and policy (and the social and economic dimensions of these) can operate effectively so that future research can be directed in the most practical and efficient manner.

**Open peer review.** To view the open peer review materials for this article, please visit http://doi.org/10.1017/ext.2023.14.

**Data availability statement.** A list of all reviewed publications is available from the corresponding author.

**Author contribution.** K.W., F.D. and F.H. were responsible for the conception and design of the research underpinning this article. C.B. wrote the first draft. All authors contributed to revised drafts and approved the final version for publication.

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

## Appendix

| | Mechanisms | Principles |
|---|---|---|
| Albert et al. (2020) | Spatial planning | Target-based laws |
| Athens (2018) | Legal Rights of Nature | |
| Artelle et al. (2019) | Indigenous Collaborations and Partnerships | |
| Ban et al. (2014) | Protected Areas | |
| Benabou (2014) | Offsets and No Net Loss Schemes | Target-based laws |
| | Climate and Biodiversity Mainstreaming | |
| Bergman et al. (2020) | | |
| Berkes and Davidson-Hunt(2006) | Indigenous Collaborations and Partnerships | Inclusive decision making |
| Bethlenfalvy and Olive (2021) | Listing and Recovery Plans | Transparent and independent decision-making and review |
| Best (2021) | | |
| Boakes et al. (2022) | Climate and Biodiversity Mainstreaming | Effective evaluation and review |
| Braby (2018) | Listing and Recovery Plans | Effective evaluation and review |
| | | Transparent and independent decision-making and review |
| Bull et al. (2013) | Offsets and No Net Loss Schemes | Transparent and independent decision-making and review |
| | | Target-based laws |
| Buxton et al. (2020) | Listing and Recovery Plans | Effective evaluation and review |
| Camaclang et al. (2015) | Listing and Recovery Plans | Threat-based laws |
| | Land Restoration | Inclusive decision making |
| Campbell et al. (2017) | | |
| Cardesa-Salzmann (2016) | | |
| Cattino and Reckien (2021) | | |
| Cresswell et al. (2021) | | |
| Dasgupta (2021) | | |
| Dee Boersma et al. (2001) | Listing and Recovery Plans | Threat-based laws |
| | | Target-based laws |
| Department of Environment and Science (Qld) (2022) | | |
| Dimitropoulou (2018) | Listing and Recovery Plans | Threat-based laws |
| Dorey and Walker (2018) | Listing and Recovery Plans | |
| Dorrough et al. (2021) | | |
| Environment and Climate Change Canada (2016) | | |
| Dudley et al. (2018) | Protected Areas | Target-based laws |
| Zu Ermgassen et al. (2019) | Offsets and No Net Loss Schemes | Target-based laws |
| | | Effective evaluation and review |
| Evans et al. (2016) | Listing and Recovery Plans | |
| Evans (2018) | Land Restoration | |
| Ferreira (2017) | Offsets and No Net Loss Schemes | Effective evaluation and review |
| Gallardo et al. (2022) | Protected Areas | |
| Fitzsimons (n.d.) | Listing and Recovery Plans | Threat-based laws |
| | | Target-based laws |
| Frantz (n.d.) | Listing and Recovery Plans | Threat-based laws |
| | | Inclusive decision making |

*(Continued)*

**Appendix**  (*Continued*)

|  | Mechanisms | Principles |
| --- | --- | --- |
| Forester and Bleby (2022) |  |  |
| Gann et al. (2019) |  |  |
| Gardner et al. (2013) | Offsets and No Net Loss Schemes | Target-based laws |
| Gelcich et al. (2017) | Offsets and No Net Loss Schemes | Inclusive decision making |
|  |  | Effective evaluation and review |
| Geldmann et al. (2015) | Protected Areas | Effective evaluation and review |
| Githiru et al. (2015) | Offsets and No Net Loss Schemes | Target-based laws |
|  | Protected Areas |  |
| Goolmeer et al. (2022a) | Indigenous Collaborations and Partnerships | Inclusive decision making |
| Goolmeer et al. (2022b) | Indigenous Collaborations and Partnerships | Inclusive decision making |
| Hao et al. (2022) |  |  |
| Henson et al. (2018) | Listing and Recovery Plans |  |
| Hilty et al. (2020) |  | Threat-based laws |
|  |  | Effective evaluation and review |
| Hoops (2022) | Legal Rights of Nature |  |
| Human Rights Committee (1994) |  |  |
| Hutchings et al. (2016) | Listing and Recovery Plans | Inclusive decision making |
|  |  | Threat-based laws |
| Kati et al. (2015) | Protected Areas | Threat-based laws |
| Invasive Species Council, & Bush Heritage Australia (2020) |  |  |
| Kearney et al. (2022) | Land Stewardship and Private Protected Areas (PPAs) | Inclusive decision making |
|  | Protected Areas |  |
| Koolen-Bourke and Peart (2021) | Indigenous Collaborations and Partnerships | Inclusive decision making |
| Jordan and Lenschow (2008) |  |  |
| Karlsson-Vinkhuyzen et al. (2017) |  |  |
| Kraus et al. (2021) | Listing and Recovery Plans |  |
|  | Climate and Biodiversity Mainstreaming |  |
| Lafferty and Hovden (2003) |  |  |
| Langpap et al. (2018) | Listing and Recovery Plans | Effective evaluation and review |
|  | Protected Areas |  |
| Larson et al. (2020) |  |  |
| Lee and Wakefield-Rann (2021) |  |  |
| Law et al. (2015) | Land Stewardship and Private Protected Areas (PPAs) | Effective evaluation and review |
|  | Land Restoration |  |
| Lee and Yan (2019) | Protected Areas | Effective evaluation and review |
| Li et al. (2020) | Listing and Recovery Plans | Threat-based laws |
|  |  | Effective evaluation and review |
| Lim (2016) |  | Effective evaluation and review |
| Lindenmayer et al. (2020) | Listing and Recovery Plans | Effective evaluation and review |
| Lockhart and Rea (2019) | Offsets and No Net Loss Schemes | Effective evaluation and review |
| Marantz and Ulibarri (2022) |  |  |

**Appendix**  (*Continued*)

| | Mechanisms | Principles |
|---|---|---|
| Mann (2015) | Offsets and No Net Loss Schemes | Effective evaluation and review |
| | | Inclusive decision making |
| Martin et al. (2020) | Protected Areas | Inclusive decision making |
| | Indigenous Collaborations and Partnerships | Effective evaluation and review |
| Martin et al. (2016) | | |
| Martin et al. (2022) | | |
| Mascia et al. (2014) | | Effective evaluation and review |
| McCormack (2018) | | |
| Mickwitz et al. (2009) | | |
| Mathieu et al. (2018) | Listing and Recovery Plans | Inclusive decision making |
| | Protected Areas | |
| McCormack (n.d.) | Climate and Biodiversity Mainstreaming | |
| McDonald et al. (2016) | Climate and Biodiversity Mainstreaming | Effective evaluation and review |
| | | Target-based laws |
| Migliorini and Wezel (2017) | Land Stewardship and Private Protected Areas (PPAs) | |
| Mitchell et al. (2018) | Land Stewardship and Private Protected Areas (PPAs) | Inclusive decision making |
| Moir and Brennan (2020) | Listing and Recovery Plans | Threat-based laws |
| No'kmaq et al. (2021) | Indigenous Collaborations and Partnerships | Inclusive decision making |
| Narain et al. (2020) | | Threat-based laws |
| | | Transparent and independent decision-making and review |
| Oberthür and Stokke (2011) | | |
| OECD (2013) | | |
| Olive and McCune (2017) | Land Stewardship and Private Protected Areas (PPAs) | Transparent and independent decision-making and review |
| | | Inclusive decision making |
| Otero et al. (2020) | | Transparent and independent decision-making and review |
| | | Effective evaluation and review |
| Pfeiffer et al. (n.d.) | Land Stewardship and Private Protected Areas (PPAs) | Inclusive decision making |
| Pirard (2012) | | Effective evaluation and review |
| Peel (2008) | | |
| Putzer et al. (2022) | | |
| Razzaque et al. (2019) | | |
| Quétier et al. (2014) | Offsets and No Net Loss Schemes | Effective evaluation and review |
| | | Transparent and independent decision-making and review |
| Ray et al. (2021) | | Effective evaluation and review |
| | | Transparent and independent decision-making and review |
| Reimerson (2013) | Indigenous Collaborations and Partnerships | Inclusive decision making |
| Richardson (2016) | | |
| Riley (2019) | | |
| Santangeli et al. (2013) | | |

(*Continued*)

**Appendix**  (*Continued*)

|  | Mechanisms | Principles |
| --- | --- | --- |
| Reside et al. (2017) | Land Stewardship and Private Protected Areas (PPAs) | Transparent and independent decision-making and review |
|  |  | Target-based laws |
| Rosin (2013) | Land Stewardship and Private Protected Areas (PPAs) | Transparent and independent decision-making and review |
|  |  | Effective evaluation and review |
| Rutherford et al. (2015) | Protected Areas |  |
| Satterfield et al. (2013) | Indigenous Collaborations and Partnerships | Inclusive decision making |
| Scheele et al. (2018) |  |  |
| Schoukens and Van Hoorick (2021) |  |  |
| Pannell et al. (2021) | Land Stewardship and Private Protected Areas (PPAs) | Inclusive decision making |
|  |  | Transparent and independent decision-making and review |
| Schuster et al. (2018) | Land Stewardship and Private Protected Areas (PPAs) |  |
| Scott (2016) | Land Stewardship and Private Protected Areas (PPAs) |  |
| Simmonds et al. (2020b) | Listing and Recovery Plans | Threat-based laws |
|  | Land Restoration | Target-based laws |
| Smith et al. (2018) | Listing and Recovery Plans | Transparent and independent decision-making and review |
|  |  | Inclusive decision making |
| Staude et al. (2020) |  | Inclusive decision making |
| Tallis et al. (2015) |  | Target-based laws |
| Tayleu et al. (2017) | Land Stewardship and Private Protected Areas (PPAs) | Target-based laws |
| Taylor et al. (2011) | Listing and Recovery Plans | Threat-based laws |
| Department of Conservation (2020) |  |  |
| Maseyk et al. (2019) | Indigenous Collaborations and Partnerships | Inclusive decision making |
| Turcotte et al. (2021) |  | Inclusive decision making |
|  |  | Effective evaluation and review |
| United Nations (n.d.-a) |  |  |
| Tsioumani (2018) |  |  |
| United Nations (n.d.-b). |  |  |
| United Nations (2009) | Offsets and No Net Loss Schemes | Effective evaluation and review |
| van Doeveren (2011) |  |  |
| Verschuuren et al. (2021) | Protected Areas | Inclusive decision making |
|  | Indigenous Collaborations and Partnerships |  |
| Walsh et al. (2013) | Listing and Recovery Plans | Threat-based laws |
|  |  | Transparent and independent decision-making and review |
| Wang et al. (2020) |  | Transparent and independent decision-making and review |
|  |  | Inclusive decision making |
| Ward et al. (2019) | Listing and Recovery Plans | Threat-based laws |

(*Continued*)

**Appendix** (*Continued*)

| | Mechanisms | Principles |
|---|---|---|
| Westwood et al. (2019) | Listing and Recovery Plans | Effective evaluation and review |
| Windle and Rolfe (2008) | | Transparent and independent decision-making and review |
| | | Effective evaluation and review |
| Woinarski et al. (2017) | | Transparent and independent decision-making and review |
| | | Inclusive decision making |
| Waltman (2016) | | |
| Wintle et al. (2019) | | |
| Total 128 articles | | |