## [Reviewer Report]

Referee report: EXT-22-0031 ‘Preventing Species Extinction: Best Practice Mechanisms in Law and Policy’

This manuscript addresses an important topic – that law should provide a critical foundation for preventing extinction. It is also very timely, given the recent (December 2022) global commitments from CoP 15, including the goal ‘Human induced extinction of known threatened species is halted’.

The manuscript approaches the issue through a review and distillation of 120 (English language) publications, and through broad application of general ‘best practice’ principles. From this approach, it distils a ‘theoretical framework’ and a set of five principles that should underpin law relating to biodiversity conservation efforts more broadly.

My misgivings about the manuscript are:

(1) It seems to overlook completely the issue of assigning legal rights to biodiversity, as one possible foundation mechanism for preventing extinction. I suspect there is a substantial body of innovative policy and law relating to this topic. As one example, in recent assessments of the impacts of the Australian Black Summer wildfires on animals, Best (2021) concluded that , the ‘property status [of animals] allows them to be treated in ways that elevate their risk during disasters. It also relegates them to a status of legal inferiority’. I think it would be useful to add a consideration of the extent to which legal rights for nature have been applied, whether these have been successful in reducing extinction risk, and how they can be applied more broadly.

(2) Likewise, there is no consideration of legal application of intergenerational equity – the obligation of our generation to leave to our descendants a world that is as biodiverse as that which we inherited. If such a principle was to be embedded in legislation, then it should substantially influence decisions taken that could otherwise lead to extinctions.

(3) There seems to be no quantitative or analytical treatment of the 120-odd references collated for this paper. There is no assessment of what seems to be working well and where, or evidence base supporting the espoused principles – e.g., in which legal settings has the rate of extinction been shown to be consequently reduced. This limited treatment is disappointing especially given that the Introduction (at line 62) recognised the need to determine ‘what is and is not working’.

(4) The principles described across pp. 8-13 describe routine biodiversity conservation considerations very generally, rather than being sharply attuned to prevention of extinction itself.

(5) While listing of species as threatened is recognised appropriately in this manuscript as a fundamental component of the armoury for preventing extinctions, there is little consideration of the pervasive weakness of such listing – that it biases against undescribed and poorly known species and groups, for which knowledge gaps subvert likelihood of listing and hence protection (Walsh et al. 2013; Taylor et al. 2018). Some consideration of ‘dark extinctions’ [the loss of a species even before it is described] (Boehm and Cronk 2021), and how law can be used to reduce their likelihood, would be warranted. The issue is especially germane given the CoP wording of its extinction commitment, that it applies to ‘known threatened species’: that is cold comfort to unknown species and those poorly known species for which knowledge shortcomings mean that they are unlikely to ever be included in lists of threatened species.

(6) At various points in the manuscript, there is appropriate recognition of the need for accountability (what person or agency is responsible if a species becomes extinct, and what should be the consequences to them of such failure?). Elsewhere, Woinarski et al. (2017) recommended that an Inquiry (akin to a coronial inquest) be mandated to assess any extinction event, to determine accountability and penalty, and to seek to remedy the shortcomings such that future extinctions are less likely. I would like to see this issue given some consideration in this manuscript.

(7) There is only a single reference given to the precautionary principle, explicitly solely in respect of offsets. I suggest it has scope for far more use in law with reference to reducing the likelihood of extinctions. For example, a recent thoroughly worked case of this was the Richards J decision of November 2022 in the Supreme Court of Victoria in relation to greater gliders and VicForests. In this case, the ruling was that timber-harvesting should be prevented, with this finding largely on the basis of the precautionary principle and the possibility that the activity would make some contribution to the likelihood of the species’ extinction.

(8) Although I recognise that the aspects of law involved in relation to extinction are very diverse, I was surprised that there is no consideration in the manuscript of biosecurity. There is a substantial legal underpinning of biosecurity, with bias largely towards agricultural impacts rather than biodiversity. However, the introduction of non-native diseases, plants and animals continues to have devastating consequences to biodiversity (especially so for Australia and other islands), and the inadequate legal settings for biosecurity was a major factor in several recent extinctions in Australia, and is likely to be the cause of many more in the near future (Fensham et al. 2020).

(9) Targets. The manuscript appropriately recognises that one necessary ingredient is the need for targets (lines 200-219). It would have been interesting and useful to know the extent to which a target of no extinctions is a key objective in laws in different countries, and the extent to which the legal settings support and require reporting on this target.

(10) Research. The Introduction foreshadowed (lines 70-73) that the overview provided in this manuscript would help ‘direct scientific research’. I could find nothing in the manuscript that followed up on this objective.

Minor considerations.

It seems a bit odd to describe the findings and conclusions within the Introduction (e.g., lines 79-95).

There are some examples of references out of alphabetical order (e.g., 407, 603).

The sole figure is not especially revealing.

Refences cited

Best A (2021) The legal status of animals: a source of their disaster vulnerability. Australian Journal of Emergency Management 36, 63-68.

Boehm MMA, Cronk QCB (2021) Dark extinction: the problem of unknown historical extinctions. Biology letters 17, 20210007.

Fensham RJ, Carnegie AJ, Laffineur B, Makinson RO, Pegg GS, Wills J (2020) Imminent extinction of Australian Myrtaceae by fungal disease. Trends in Ecology & Evolution 35, 554-557.

Taylor GS, Braby MF, Moir ML, Harvey MS, Sands DPA, New TR, Kitching RL, McQuillan PB, Hogendoorn K, Glatz RV, Andren M, Cook JM, Henry SC, Valenzuela I, Weinstein P (2018) Strategic national approach for improving the conservation management of insects and allied invertebrates in Australia. Austral Entomology 57, 124-149.

Walsh JC, Watson JEM, Bottrill MC, Joseph LN, Possingham HP (2013) Trends and biases in the listing and recovery planning for threatened species: an Australian case study. Oryx 47, 134-143.

Woinarski JCZ, Garnett ST, Legge SM, Lindenmayer DB (2017) The contribution of policy, law, management, research, and advocacy failings to the recent extinctions of three Australian vertebrate species. Conservation Biology 31, 13-23.

---

## [Reviewer Report]

This article aims to determine best practice mechanism (and principles) in preventing species extinctions. The scope and scale of this objective is quite daunting, so I appreciate the difficulty the authors had in executing this task. The resulting best practice principles and mechanisms, summarised in Figure 1, are nothing particularly new or surprising – which is probably fine for a Review article. 

My main concern with the piece is its lack of methodological rigor. The authors conduct a literature review, using a set of keywords and search two online databases. The inclusion/exclusion criteria are fairly vague, and there’s scant details of how judgements were made and resolved in a multi-author team. There’s no details of how many articles were identified initially, excluded via criteria, and added via snowball (as per PRISMA guidelines https://prisma-statement.org/). Now the authors conduct a qualitative review, so a strict quantitative method doesn’t need to be followed – but some detail would be nice, and unfortunately the qualitative rigor is also low.

The authors indicate they use a “manual thematic analysis of the literature” based on principles of good governance (accountability, transparency, inclusiveness, participation, efficiency and effectiveness, rule of law; van Doeveren (2011)). The authors then somehow use these good governance principles to identify best practice principles and mechanisms for biodiversity conservation and law. There are no details on qualitative methodology or method (e.g see Moon et al. 2019). There are no details on the “manual thematic analysis” method – did they adopt a specific style of coding, or was it a general vibe? I include a list of potential methodological resources the authors could draw on at the end of my review.

The unclear coding method and unclear lit review method combines to cast doubt on exactly why and how the authors determined their best practice principle and mechanisms. E.g line 303 “One prominent theme that emerged from the literature “. How was this identified as a prominent theme? Was it the relative strength of the theme (qualitative coding/thematic analysis) or the number of codes to indicate strength (quantitative content analysis)? Further, because the authors included both academic literature and grey literature (e.g NGO reports) it’s not possible to determine the validity of the strength of this theme. E.g if strength was determined according to a bunch of NGO reports saying something, this could be less valid than a bunch of academic papers saying something. The authors do not provide their list of 120 articles contained in their final database or any details about their geographic coverage (apart from indicating they largely cover Western legal systems) or which specific countries/jurisdictions are covered. 

One glaring omission from the list of best practice principles/mechanisms for me was adequate resourcing. E.g in Australia, a series of Australian National Audit Office reports highlight the lack of resources made available to enable EPBC Act implementation. I’m sure the OECD highlights lack of resourcing to implement environmental laws in a recent report too. Another issue that occurs in Australia, and likely elsewhere, is ambiguity and discretion. It would be good to understand why these issues were not picked up by the review, and if it’s because they’re not important or if it’s an omission due to method. 

Some more detailed comments below.

Line 32: “Biodiversity conservation law” - is this a term you are defining or are citing from elsewhere? If the former, make this explicit.

Line 48: both Woinarski 2016 and 2017 are referred to, but only 2017 is in reference list

Line 56: “Even laws drafted to specifically address species extinction….(see for example Woinarski et al. 2016)” - this seems to be referring to the EPBC Act - which has an object relating to conserving biodiversity, but not specifically about extinction, so suggest revising sentence. I think the USA endangered species act may have specific reference to extinction.

Line 181: “threat levels” - is this referring to conservation status? i.e vulnerable, endangered etc?

Line 261 and line 292: “Commentators” - commentators? or authors/scholars? how much of the literature was academic and how much NGO/grey?

Line 304: “mainstreaming” - may also be referred to as horizontal policy integration in the policy literature

Line 361: “Empirical evidence has suggested that protected areas provide the most effective legal mechanism for large-scale biodiversity conservation (Evans, 2016).” - this is an incredible claim, and the reference cited does not provide this evidence

Line 409: “Walsh et al. argued…” - doesn’t this assume there is a federal biodiversity law in addition to/instead of subnational biodiversity laws?

Line 449: unsure why the Reside reference is used to support a statement about private land conservation. Pannell et al. 2017 not included on reference list

Line 447: Land Restoration – need to make explicit that you seem to be considering biodiversity conservation in the terrestrial, not marine or freshwater realm?

Suggested references

Dixon-Woods, M., Agarwal, S., Jones, D., Young, B., Sutton, A., 2005. Synthesising qualitative and quantitative evidence: A review of possible methods. J Health Serv Res Policy 10, 45–53. https://doi.org/10.1177/135581960501000110 

Hsieh, H.-F., Shannon, S.E., 2005. Three Approaches to Qualitative Content Analysis. Qual Health Res 15, 1277–1288. https://doi.org/10.1177/1049732305276687 

Munn, Z., Peters, M.D.J., Stern, C., Tufanaru, C., McArthur, A., Aromataris, E., 2018. Systematic review or scoping review? Guidance for authors when choosing between a systematic or scoping review approach. BMC Medical Research Methodology 18, 143. https://doi.org/10.1186/s12874-018-0611-x

Moon, K., Blackman, D.A., Adams, V.M., Colvin, R.M., Davila, F., Evans, M.C., Januchowski‐Hartley, S.R., Bennett, N.J., Dickinson, H., Sandbrook, C., Sherren, K., John, F.A.V.S., Kerkhoff, L. van, Wyborn, C., 2019. Expanding the role of social science in conservation through an engagement with philosophy, methodology, and methods. Methods in Ecology and Evolution 10, 294–302. https://doi.org/10.1111/2041-210X.13126

Moon, K., Brewer, T.D., Januchowski-Hartley, S.R., Adams, V.M., Blackman, D.A., 2016. A guideline to improve qualitative social science publishing in ecology and conservation journals. Ecology and Society 21. https://doi.org/10.5751/ES-08663-210317

Vaismoradi, M., Turunen, H., Bondas, T., 2013. Content analysis and thematic analysis: Implications for conducting a qualitative descriptive study. Nursing & Health Sciences 15, 398–405. https://doi.org/10.1111/nhs.12048

---

## [Reviewer Report]

Referee report: EXT-22-0031.R1 Preventing species extinction: Best practice mechanisms in law and policy.

I reviewed the original submission of this study. This revised version is improved: it has clarified some methodological and interpretational issues, and added some text on matters that were missing or underdone in the original submission.

However, there still seems to be a fundamental mis-match between the objectives (as indicated in the ‘preventing species extinctions’ title) and the approach of the study and interpretation of its results. This study is essentially a trawl through published papers on conservation law, from which some common mechanisms and principles are distilled. My major concerns remain: 

(i) there is little in the study explicitly about preventing extinction; 

(ii) the justification of ‘best practice’ (mechanisms and principles) is obscure; 

(iii) there is no linking of the putative ‘best practice’ mechanisms or principles with an evidence base that these mechanisms or principles affect or determine rates of success or failure in preventing extinction. This lack of presented evidence in relation to extinction outcomes is notwithstanding the explicit claim (lines 11-12) that the paper would highlight linkages to success; and

(iv) there are fatal omissions. 

There seems to be retro-fitting of the study to the objective of ‘preventing species extinctions’.

For example, one concern I raised in my previous review was the lack of consideration of biosecurity law. This omission is now referenced in the revision at lines 113, 129, 600-602, with the belated rationale now that ‘it was outside the analysis of this project’ (line 602). Biosecurity law and practice is critical for preventing extinction. 75% of vertebrate species extinctions have been of island species, along with vast numbers of invertebrate extinctions, and 40% of threatened species are island endemics (Régnier et al. 2015a; Régnier et al. 2015b; McCreless et al. 2016). The driving factor of these extinctions has been, and continues to be, the introduction of pest and weed species; and the mechanism to address this threat is biosecurity law. Any review of legislation relating to extinction will be inadequate if this issue is not appropriately considered.

My previous review also noted that legal treatment or inclusion of the principle of intergeneration equity and the precautionary principle was also likely to be foundational for preventing extinction. The omission of these from this study is now noted (at lines 169-171), with the rationale that instead ‘we focus specifically on those principles relevant to biodiversity conservation’. This logic seems wayward given the non-specificity of other principles that are included, such as transparency, inclusivity and review (e.g., lines 167-168). Again, I think the treatment in this submission falls short of the standard required in a review of legal options and mechanisms relevant to preventing species extinction.

One concern I had in my review of the original submission was the lack of consideration of legal rights for biodiversity. The authors have done a good job (lines 540-576) in now rectifying that omission

Minor comments

Lines 21-27. The Abstract would be more informative if it explicitly described the conclusions relating to best practice, rather than vaguely state that these matters will be explored.

Lines 64-65 (and possibly elsewhere). The ordering of references appears haphazard.

Lines 329-338. I like the discussion around ensuring prevention of species extinction as a core priority of law

Lines 483-484: improve on ‘Due to the limited nature imposed by the ‘voluntary’ nature of these mechanisms ...’

Line 519. Replace ‘predominately’ with ‘predominantly’

References cited

McCreless EE, Huff DD, Croll DA, Tershy BR, Spatz DR, Holmes ND, Butchart SH, Wilcox C (2016) Past and estimated future impact of invasive alien mammals on insular threatened vertebrate populations. Nature Communications 7.

Régnier C, Achaz G, Lambert A, Cowie RH, Bouchet P, Fontaine B (2015a) Mass extinction in poorly known taxa. Proceedings of the National Academy of Sciences 112, 7761-7766.

Régnier C, Bouchet P, Hayes KA, Yeung NW, Christensen CC, Chung DJD, Fontaine B, Cowie RH (2015b) Extinction in a hyperdiverse endemic Hawaiian land snail family and implications for the underestimation of invertebrate extinction. Conservation Biology 29, 1715-1723.

---

## [Reviewer Report]

Thank-you for providing me the opportunity to review this manuscript again. I think it is much improved following the authors revisions. I have a few final comments on the revised manuscript.

I was going to say that, despite the author’s inclusion of a clear definition of what they mean by best practice, and good discussion of how best practice doesn’t necessarily mean effective, I’ve still been a bit unsure about the term. I was more assured when I realised best practice was included explicitly in the search string – suggest making a brief explicit flag to this. But the title of “Preventing Species Extinction: Best Practice Mechanisms in Law and Policy” isn’t really consistent with the author’s finding that “the concept of ‘best practice’ law and policy for mechanisms to prevent species extinction was elusive in the literature” and “does not necessarily equate to a reversal in species decline” and “Further, most western countries utilise the same legal mechanisms (also known as policy tools) to reduce extinction”. Given these findings, I’d suggest just shortening the title to “Best Practice Mechanisms for Biodiversity Conservation Law and Policy” or something that points to the ambiguity in the term, or the broad scope of how you define biodiversity conservation law in lines 36 to 105. 

Line 693: “successful threat abatement plans have reduced the deaths of albatross and petrels in Australia.” – there’s no citation provided with evidence to support this statement. There’s also likely conflation between the presence of the instrument (threat abatement plan) and funding available to implement the plan. For example Bottrill et al. 2011 found there was no difference in recovery outcomes for species with or without a recovery plan in Australia – they were unable to assess funding data but this is likely a major contributor. 

Line 809: “objective/target” note that in the conservation planning literature, an objective is an overall goal – e.g to prevent species extinctions, whereas a target is a quantitative measure put in place to help evaluate progress, e,g to protect 30% land area – these are two different things. I suggest just clarifying and potentially simplifying the sentence from lines 807 to 811 as it’s quite long. My feeling is you’re referring more to progress against targets. Generally there’s a few too many forward slashes in the manuscript, suggest replacing with “or” as much as possible if that’s the intended meaning. 

Line 940: “Empirical evidence has suggested that protected areas can provide an effective legal mechanism for biodiversity conservation: for example, a study monitored populations of manatee in Florida before and after an area was closed to boating, and it found fewer manatee were killed once the area was protected (Evans et al, 2016).”

I’m actually not sure this specific example is that helpful here. The general idea is that an effective protected area deliver an outcome that wouldn’t have occurred in absence of the protection mechanism – that is, there is additionality. i’m guessing if your search was broader you would have gotten into the paper parks/protected area evaluation literature discussion, for example work by Geldmann, Barnes, Rodrigues. I understand that’s out of your scope, but there’s been a huge amount of work evaluating protected area effectiveness at global and national scales using quasi-experimental designs and satellite imagery, that is somewhat trivialised the single manatee in Florida example. So suggest bringing it back to the general idea around additionality. 

Line 1066: My comment on the original manuscript referring to land restoration occurring in terrestrial areas may have been unclear. What I meant is, it is not until this section (but actually, the previous section on Land Stewardship) that there is a suggestion that your review is focussing on biodiversity conservation law in the terrestrial realm only. Protected areas and restoration can occur in both the terrestrial and marine realm, but you refer specifically to “land restoration here”. But note you refer to a seabird example early in the manuscript. So I actually think you’ve been fairly agnostic over whether you’re interested in terrestrial, marine or freshwater realms – so I would instead just change “Land restoration” to ecosystem restoration, and perhaps just stewardship instead of land stewardship, or potentially land and sea stewardship. Perhaps you might have gotten more marine results if you added “overexploitation” to your search strings. Whatever your choice, just be clear about your scope and be consistent with terminology.

---

## [Editor Report]

Both reviewers are concerned that the title and promise of the paper ‘preventing species extinctions’ is mismatched from the mode and scope of the research. I concur and suggest better matching and narrowing the promise in the framing. This referee makes a compelling case that invasive species and hence biosecurity law is a significant omission, as are intergenerational equity and the precautionary principle. Please reconsider and / or make the case that these are significant ’future directions’ beyond the scope of this review. Both reviewers make additional important points of clarification that the authors would do well to pay attention to in their review.

---

## [Reviewer Report]

Referee report: EXT-22-0031.R2 ‘Best practice mechanisms for biodiversity conservation law and policy’

I reviewed two previous iterations of this manuscript. To a reasonable extent, the current version has now addressed my major concerns. This draft acknowledges that more consideration of biosecurity law, intergenerational equity and the precautionary principle is warranted, but not readily incorporated into the existing study.

I still have some qualms about how ‘best practice’ has been defined. The abstract notes that ‘The purpose of this paper was to evaluate ... legal systems that can be considered successful, or unsuccessful’. However, the paper provides no robust evidence about success or failure, or assessment of the individual and collective contribution of the derived principles and mechanisms to reducing the rate or likelihood of extinction or contributing to biodiversity recovery. Instead, self-identified ‘best practice’ seems to be treated as success, and all principles and mechanisms appear to be treated as equally important. There is also no consideration of the overall extent to which these best practice principles and mechanisms are actually represented in law. It would be useful to include an additional sentence or two on these issues in the introduction or discussion.

Minor comment:

Line 56. Change ‘commonly biodiversity loss’ to ‘commonly cause biodiversity loss’

---

## [Editor Report]

The reviewers make the final point that needs to be addressed, and that is of the definition of success, and in this MS it is self-defined. This is a low-bar and should be acknowledged as such. Please dd the sentences suggested.

“However, the paper provides no robust evidence about success or failure, or assessment of the individual and collective contribution of the derived principles and mechanisms to reducing the rate or likelihood of extinction or contributing to biodiversity recovery. Instead, self-identified ‘best practice’ seems to be treated as success, and all principles and mechanisms appear to be treated as equally important. There is also no consideration of the overall extent to which these best practice principles and mechanisms are actually represented in law. It would be useful to include an additional sentence or two on these issues in the introduction or discussion.”